# Development of Machine Learning Models to Predict Platinum Sensitivity of High-Grade Serous Ovarian Carcinoma

**DOI:** 10.3390/cancers13081875

**Published:** 2021-04-14

**Authors:** Suhyun Hwangbo, Se Ik Kim, Ju-Hyun Kim, Kyung Jin Eoh, Chanhee Lee, Young Tae Kim, Dae-Shik Suh, Taesung Park, Yong Sang Song

**Affiliations:** 1Interdisciplinary Program in Bioinformatics, Seoul National University, Seoul 08826, Korea; tngus8695@snu.ac.kr (S.H.); lch951022@snu.ac.kr (C.L.); 2Department of Obstetrics and Gynecology, Seoul National University College of Medicine, Seoul 03080, Korea; seikky@naver.com; 3Department of Obstetrics and Gynecology, Graduate School of Medicine, University of Ulsan, Seoul 05505, Korea; smilekako@naver.com; 4Department of Obstetrics and Gynecology, Yongin Severance Hospital, Yonsei University College of Medicine, Yongin-si 17046, Korea; kjeoh2030@yuhs.ac; 5Department of Obstetrics and Gynecology, Institute of Women’s Life Medical Science, Yonsei University College of Medicine, Seoul 03722, Korea; ytkchoi@yuhs.ac; 6Department of Obstetrics and Gynecology, Asan Medical Center, University of Ulsan College of Medicine, Seoul 05505, Korea; dssuh@amc.seoul.kr; 7Department of Statistics, Seoul National University, Seoul 08826, Korea; 8Cancer Research Institute, Seoul National University College of Medicine, Seoul 03080, Korea

**Keywords:** ovarian cancer, high-grade serous carcinoma, prognosis, platinum resistance, machine learning, model

## Abstract

**Simple Summary:**

High-grade serous ovarian carcinoma (HGSOC) is the most aggressive histologic type of epithelial ovarian cancer, associated with high recurrence and mortality rates despite standard treatment. In accordance with the era of precision cancer medicine, we aimed to develop machine learning models predicting platinum sensitivity in patients with HGSOC. First, we collected patients’ clinicopathologic data from three tertiary hospitals. Second, we elected six variables associated with platinum sensitivity using the stepwise selection method. Third, based on these variables, predictive models were constructed using four machine learning algorithms, logistic regression (LR), random forest, support vector machine, and deep neural network. Evaluation of model performance with the five-fold cross-validation method identified the LR-based model as the best at identifying platinum-resistant cases. Lastly, we developed a web-based nomogram by fitting the LR model results for clinical utility. Based on the prediction results, physicians may implement individualized treatment and surveillance plans for each HGSOC patient.

**Abstract:**

To support the implementation of individualized disease management, we aimed to develop machine learning models predicting platinum sensitivity in patients with high-grade serous ovarian carcinoma (HGSOC). We reviewed the medical records of 1002 eligible patients. Patients’ clinicopathologic characteristics, surgical findings, details of chemotherapy, treatment response, and survival outcomes were collected. Using the stepwise selection method, based on the area under the receiver operating characteristic curve (AUC) values, six variables associated with platinum sensitivity were selected: age, initial serum CA-125 levels, neoadjuvant chemotherapy, pelvic lymph node status, involvement of pelvic tissue other than the uterus and tubes, and involvement of the small bowel and mesentery. Based on these variables, predictive models were constructed using four machine learning algorithms, logistic regression (LR), random forest, support vector machine, and deep neural network; the model performance was evaluated with the five-fold cross-validation method. The LR-based model performed best at identifying platinum-resistant cases with an AUC of 0.741. Adding the FIGO stage and residual tumor size after debulking surgery did not improve model performance. Based on the six-variable LR model, we also developed a web-based nomogram. The presented models may be useful in clinical practice and research.

## 1. Introduction

Ovarian cancer accounted for approximately 313,959 new cases and 207,252 deaths in 2020, ranking eighth in both incidence and mortality among female cancers, globally [1]. Ovarian cancer epidemiology shows regional differences according to the level of development [2]. In the United States, ovarian cancer accounts for 4.9% of female cancer-related deaths, ranking the disease as the fifth most deadly female cancer [3]. In Korea, ovarian cancer incidence is rapidly increasing, likely due to population aging and the uptake of sedentary lifestyles [4]. In the absence of cancer-specific symptoms and effective screening tools, epithelial ovarian cancer tends to be diagnosed at an advanced stage, leading to high recurrence and mortality rates despite treatment [5].

Approximately 90% of ovarian cancers are epithelial; high-grade serous ovarian carcinoma (HGSOC) is the most common and aggressive histologic ovarian cancer type [6]. Presently, the primary treatment for HGSOC is cytoreductive surgery followed by platinum-based combination chemotherapy [7,8,9]. Nevertheless, most HGSOC patients undergoing primary treatment are at high risk of recurrence due to chemoresistance [10]. In general, patients are divided into two groups according to the duration of a platinum-free interval (PFI), which is the time interval from the completion of platinum-based chemotherapy to disease progression [11]. Patients with PFI of <6 months are considered “platinum-resistant”, have poor prognosis with a median survival of <12 months, and a response rate of <15% to subsequent chemotherapy [12].

In the era of precision cancer medicine, predicting platinum sensitivity with high accuracy remains a challenge. A patient likely to be platinum-resistant might undergo more aggressive treatment in addition to the standard primary treatment. For example, angiogenesis inhibitors or poly(ADP-ribose) polymerase (PARP) inhibitors may be added as maintenance therapy after the completion of chemotherapy [13,14,15]. Conventional intraperitoneal chemotherapy or hyperthermic intraperitoneal chemotherapy may be considered in this group [16]. In addition, patients may undergo intensive recurrence surveillance with an adjusted checkup schedule, including measuring serum CA-125 or undergoing imaging studies more frequently. However, among previously developed prognostic prediction models or nomograms for the primary treatment of ovarian cancer, only a few can predict platinum sensitivity [17], as most previous studies have focused on the prediction of progression-free (PFS) [18] or overall survival (OS) [19,20], or both [21,22].

In addition, few studies have developed prognostic prediction models based on a specific histologic type of epithelial ovarian cancer; most of the previous studies included HGSOC alongside other histologic types in the same study population. Previously, our research team developed a nomogram predicting platinum sensitivity using clinicopathologic data of 710 patients with epithelial ovarian cancer, including 389 (54.8%) with HGSOC [22]. In that study, the model showed a performance identifying platinum-resistant cases with the area under the receiver operating characteristic (ROC) curve (AUC) of 0.758. However, applying this model to an extended cohort, we observed a significant drop in the predictive performance. Considering that most multi-omics studies on ovarian cancer have been conducted in HGSOC, it is necessary to focus on the HGSOC and develop models predicting platinum sensitivity in this patient group. Moreover, if we apply the machine learning method, a data analytics technique emerging in the biomedical research field, we think it will be possible to develop models with higher predictive capabilities.

Thus, we aimed to develop machine learning models predicting platinum sensitivity in patients with HGSOC to support the implementation of individualized treatment and surveillance in this population. Patients’ clinicopathologic data were collected from three tertiary hospitals, and various machine learning algorithms were used.

## 2. Materials and Methods

### 2.1. Study Population and Data Collection

From the Ovarian Cancer Cohort Database of each institution, we identified patients who met the following inclusion criteria: (1) aged ≥ 19 years; (2) pathologically confirmed HGSOCs; (3) diagnosed between January 2000 and June 2019; and (4) underwent primary treatment, either primary debulking surgery (PDS) followed by platinum-based postoperative adjuvant chemotherapy or platinum-based neoadjuvant chemotherapy (NAC) followed by interval debulking surgery and postoperative adjuvant chemotherapy. Meanwhile, patients were excluded if they met any of the following criteria: (1) were immunocompromised or pregnant; (2) did not receive platinum-based combination chemotherapy; (3) had synchronous double primary cancers; and (4) were lost to follow-up before reaching six months of PFI without evidence of disease recurrence, precluding the determination of platinum sensitivity. In addition, patients who received front-line chemotherapy containing bevacizumab or PARP inhibitor maintenance therapy were also excluded for fair comparisons of platinum sensitivity.

We extracted clinicopathologic data from patients’ medical records and pathologic reports. Detailed information on the type of collected variables was the same as our previous study [22] except for differential blood cell counts at initial diagnosis, which were not collected in this study.

The three institutions are conducting NAC on a similar basis. In general, we apply NAC when the patients meet one of the following conditions: (1) high tumor burden on initial imaging studies with multiple and unresectable extra-abdominal metastases, multiple liver parenchymal metastases or pulmonary metastases, and extensive small bowel/mesenteric root involvement; (2) poor performance status and high operative risk with severe comorbidities; or (3) if suboptimal primary cytoreduction is expected (residual disease measuring >1 cm).

At all participating institutions, patients underwent surveillance, consisting of computed tomography (CT) scans and serum CA-125 level measurements every three cycles of chemotherapy, and every three months for the first year after the completion of primary treatment, and every six months for three years thereafter. Disease progression or recurrence was ascertained based on CT scans, using the Response Evaluation Criteria in Solid Tumors (RECIST) version 1.1 in patients with measurable disease [23] or based on the serum CA-125 levels, using Gynecologic Cancer InterGroup criteria in patients with unmeasurable disease [24].

For the present study, patients who experienced disease recurrence with PFI of <6 months were assigned to the platinum-resistant group, while those with PFI ≥6 months, regardless of disease recurrence status, were assigned to the platinum-sensitive group. In addition to platinum sensitivity, which was the primary endpoint of this study, we collected patients’ survival data. PFS was defined as the time interval between the start date of primary treatment and the date of disease progression, while OS was defined as the time interval between the date of initial diagnosis and the date of cancer-related death or the end of the study period.

### 2.2. Exploratory Data Analysis

Among the variables collected by the participating institutions, 42 were considered independent variables and underwent exploratory analysis (Appendix A). Specifically, we identified the distribution of each continuous variable and created a contingency table for each categorical variable. Variables with skewed distributions (e.g., serum CA-125 levels) were ln-transformed to correct the skewness. To identify candidate markers related to platinum sensitivity, we performed univariate analysis. The Wilcoxon rank-sum test, chi-square test, and log-rank test were used for continuous, binary, and time-to-event variables, respectively. For variables with more than three categories, we performed a likelihood ratio test by comparing the deviance of the full model (FM: $\mathrm{logit}\left(p\left(\mathrm{Resistant}\right)\right)={\;\beta }_{0}+\beta_{1}\mathrm{Variable}$) with that of the reduced model (RM: $\mathrm{logit}\left(p\left(\mathrm{Resistant}\right)\right)={\;\beta }_{0}$).

### 2.3. Variable Selection

To construct a model that predicts platinum sensitivity, we used stepwise selection based on the AUC values. This AUC-based stepwise variable selection method was also used in our previously published studies [22,25]. The leave-one-out cross-validation (LOOCV) method was used to calculate the AUC value for the validation set (${\mathrm{AUC}}_{\mathrm{LOOCV}}$). The detailed process of the model selection was as follows:

(1) Specify the initial model $M_{0}$ with no predictors. Set t = 0 and set ${\mathrm{AUC}}_{\mathrm{LOOCV}}=0.5$. 

(2) Forward step: Let t = t + 1. For each predictor X that is not included in $M_{t-1}$, perform steps (2-1) to (2-4) and select the model $M_{F}$ with the highest ${\mathrm{AUC}}_{\mathrm{LOOCV}}$. If $M_{F}$ > $M_{t-1}$, update the model $M_{t}$ to be $M_{F}$. Otherwise, $M_{t}={\;M}_{t-1}$, stop the algorithm and report $M_{t}$ as the final model.

(2-1) Fit the model $\left\{M_{t-1},\;X\right\}$ using n-1 training samples.

(2-2) Calculate the predicted probability of a validation sample by applying the fitted model to the validation sample.

(2-3) Iterate step (2-1) and (2-2) until the predicted probabilities for all validation samples are calculated.

(2-4) Calculate AUC using n predicted probabilities and true labels (i.e., ${\mathrm{AUC}}_{\mathrm{LOOCV}}$).

(3) Backward step: Let t = t + 1. For each predictor X that is included in $M_{t-1}$, perform step (3-1) to (3-4) and select the model $M_{B}$ with the highest ${\mathrm{AUC}}_{\mathrm{LOOCV}}$. If $M_{B}$ > $M_{t-1}$, update the model $M_{t}$ to be $M_{B}$. Otherwise, $M_{t}={\;M}_{t-1}$ and proceed to step (2).

(3-1) Fit the model $\left\{M_{t-1},-X\right\}$ using n-1 training samples.

(3-2) Calculate the predicted probability of a validation sample by applying the fitted model to the validation sample.

(3-3) Iterate step (3-1) and (3-2) until the predicted probabilities for all validation samples are calculated.

(3-4) Calculate AUC using n predicted probabilities and true labels (i.e., ${\mathrm{AUC}}_{\mathrm{LOOCV}}$).

### 2.4. Model Development and Validation

We developed predictive models with selected variables and evaluated the performance of these models. Four machine learning algorithms were used: logistic regression (LR), random forest (RF) [26], support vector machine (SVM) [27], and deep neural network (DNN) [28]. A list of machine-learning methods is presented in Appendix A. We compared the performance metrics among the models using the five-fold cross-validation method. Splitting the whole dataset for five-fold cross-validation, we considered the proportions of platinum-resistant cases and institutions to reduce the heterogeneity. As the metrics of interest, we used AUC, sensitivity, specificity, and balanced accuracy estimates. To present sensitivity and specificity, we chose a threshold value with the maximum balanced accuracy. For RF, SVM, and DNN, we tuned hyperparameters to get the optimal hyperparameter combination with the highest mean of validation AUC values.

### 2.5. Statistical Analysis

Differences in the clinicopathologic characteristics were evaluated between the platinum-sensitive and resistant groups. We used the Kaplan–Meier methods with log-rank test for survival analysis. Survival analyses were performed using IBM SPSS Statistics software (version 25.0; SPSS Inc., Chicago, IL, USA), while all other statistical analyses were performed using R statistical software version 3.6.1 (R Foundation for Statistical Computing, Vienna, Austria; http://www.R-project.org, accessed on 1 December 2020). A *p* value < 0.05 was considered statistically significant.

## 3. Results

### 3.1. Characteristics of the Study Population

The overall study design is presented in Figure 1. In total, 1002 patients were included in this study: 568 (56.7%), 246 (24.6%), and 188 (18.8%) from the Seoul National University Hospital, Asan Medical Center, and Severance Hospital, respectively. Among the study population, 388 (38.7%) were also part of the study population assessed in our previous study [22]. In terms of primary treatment, 764 (76.2%) patients underwent PDS, while 238 (23.8%) patients received NAC followed by interval debulking surgery. Among the study population, 223 (22.3%) and 779 (77.7%) patients were assigned to the platinum-resistant and -sensitive groups, respectively. The distribution of the platinum-sensitive and -resistant patients was similar among the participating institutions (*p* = 0.164; Appendix A).

During the median observation period of 41.5 (range, 4.0 to 224.0) months, 734 (73.3%) and 305 (30.4%) patients experienced disease recurrence and died, respectively. The median PFS and OS estimates for all patients were 19.2 and 119.8 months, respectively (Figure 2A,B). Compared to the platinum-sensitive group, the platinum-resistant group showed significantly worse PFS (median, 9.2 vs. 25.1 months; *p* < 0.001) and OS (median, 30.6 vs. 144.8; *p* < 0.001) (Figure 2C,D).

Patients’ clinicopathologic characteristics are presented in Table 1. The platinum-resistant group was significantly older (mean, 58.3 vs. 55.2 years; *p* < 0.001), and had a higher proportion of International Federation of Gynecology and Obstetrics (FIGO) stage III-IV disease (99.1% vs. 87.9%; *p* < 0.001), and higher serum CA-125 levels (mean ln-transformed value, 7.1 vs. 6.6 IU/mL; *p* < 0.001) than did the platinum-sensitive group. The proportion of NAC recipients was higher in the platinum-resistant group (39.5% vs. 19.3%; *p* < 0.001) than in the platinum-sensitive group. After cytoreductive surgery (PDS or interval debulking surgery after NAC), complete cytoreduction was less likely to be achieved in the platinum-resistant group than in the platinum-sensitive group (43.9% vs. 61.2%; *p* < 0.001). The frontline chemotherapy regimen was similar in both groups (*p* = 0.870); however, the total number of frontline chemotherapy cycles was different (*p* < 0.001).

Details of surgical procedures and the associated findings are shown in Appendix A. There was no difference in the rates of lymph node (LN) dissection, large bowel resection, or upper abdominal surgery between the groups. However, the groups showed significant differences in the rates of pelvic LN metastasis (*p* < 0.001), para-aortic LN metastasis (*p* = 0.002), tumor involvement of the small bowel and mesentery (*p* < 0.001), tumor involvement of the colon other than rectosigmoid (*p* = 0.038), tumor involvement of the diaphragm (*p* < 0.001), liver parenchyma metastasis (*p* = 0.019), and pleural effusion (*p* < 0.001).

### 3.2. Model Development and Validation

Through the variable selection step, the following six variables were selected: age (continuous), serum CA-125 levels (ln-transformed, continuous), primary treatment strategy (NAC vs. PDS), pelvic LN status (metastasis vs. no metastasis), tumor involvement of pelvic tissue other than uterus and tube (macroscopic vs. microscopic vs. no involvement), and tumor involvement of the small bowel and mesentery (>2 cm vs. ≤2 cm vs. microscopic or no involvement). 

Based on these variables, we developed machine learning models predicting platinum sensitivity, using LR, RF, SVM, and DNN methods. Table 2 presents each model’s performance identifying platinum-resistant cases. The four models’ ROC curves, created from the five-fold cross validation, are shown in Figure 3A. Among them, the LR-based model showed the best performance identifying platinum-resistant cases with an AUC of 0.741 (sensitivity, 0.778; specificity, 0.622; balanced accuracy, 0.700 at the cut-off value of 0.175). We further added one or two of the well-known prognostic factors, FIGO stage and residual tumor size after debulking surgery, and developed machine learning models. However, performance of the seven or eight-variable models was similar to those of the six-variable model (Table 2 and Figure 3B–D).

Next, we compared patients’ survival outcomes between platinum-sensitive and -resistant groups predicted by the six-variable, LR model with a cut-off value of 0.175 (Appendix A). Regardless of the validation set, the platinum-resistant group showed significantly worse PFS and OS than the platinum-sensitive group. These results suggest that the developed predictive model also discriminates patients with good and poor survival outcomes well.

Finally, we developed a nomogram based on the six-variable LR model. Appendix A presents fitted results of the LR model used for nomogram development. The developed nomogram presents total points as well as the probability of being platinum-resistant cases. Based on the cut-off value of 41 points, we regarded the case with total point ≥ 41 as a high-risk group. A user-friendly interface was implemented for the developed nomogram and posted on a website to facilitate clinical use (http://statgen.snu.ac.kr/software/nomogramHGSOC/, accessed on 15 January 2021). In this web-based nomogram, the input of risk factors and output of calculated results are operated by HTML and CGI files, respectively (Figure 4). 

## 4. Discussion

In the current study, we successfully developed machine learning models predicting platinum sensitivity after primary treatment in patients with HGSOC. Based on six systematically selected variables, predictive models were developed using LR, RF, SVM, and DNN methods; among them, the LR-based model performed best at identifying platinum-resistant cases, with the AUC of 0.741. For clinical purposes, we also developed a web-based nomogram by fitting the LR model results. To our knowledge, this study is the first that applied machine learning algorithms in model development for the prediction of HGSOC patients’ platinum sensitivity.

This study was based on a hypothesis that predictive models targeting HGSOC may differ from those targeting epithelial ovarian cancer of all histological types. To the best of our knowledge, this study is the first to present machine learning models that predict platinum sensitivity in patients with HGSOC. Previous studies aimed at developing predictive models of platinum sensitivity did not confine ovarian cancer patients to a specific histologic type, such as HGSOC [17,22]. While Paik et al.’s study included only PDS cases [17], our previous and current studies put PDS and NAC cases together in the model development [22]. The selected variables also differed among the models [17,22]. Such differences in study population and selected variables make it difficult to conduct comparisons of predictive models, even in the same population. 

Nevertheless, we could compare the predictive performance of models in the current study population by excluding samples with missing values (complete dataset) or generating imputed datasets using multivariate imputation by the chained equations algorithm. As a result, the six-variable LR model of the current study had significantly superior predictive power than our previous model (AUCs, 0.762 vs. 0.667; *p* = 0.009 in the complete dataset; 0.706 vs. 0.639; *p* < 0.05 in the imputed dataset), but similar to Paik et al.’s model. As the origin and molecular pathogenesis of HGSOC differ from those of the other histologic types, translating to distinct clinical features [29,30,31], it follows that predictive models should be dedicated to specific histologic types.

During the AUC-based stepwise selection, where all 42 independent variables were evaluated fairly without giving any priority to the specific variables, the six variables were selected. In contrast, both the FIGO stage and residual tumor size after debulking surgery, which are the best-known prognostic factors in ovarian cancer, were not selected. We added either the FIGO stage or residual tumor size after debulking surgery, or both, and developed machine learning models. However, the seven or eight-variable models did not perform better at identifying platinum-resistant cases than did the original six-variable models. 

If the FIGO stage and residual tumor size were essential variables for platinum sensitivity, they would have been identified as such in the variable selection step, which they were not. At the same time, we cannot help but point out that among the six selected variables, “pelvic LN status”, “involvement of pelvic tissue other than uterus and tube”, and “tumor involvement of the small bowel and mesentery” are directly associated with FIGO stage. In particular, the latter is also associated with residual tumor size as multiple tumors in the small bowel and mesentery are difficult to remove completely. The combination of the selected variables seems to be more suitable for reflecting disease status and consequently shows better predictive performance than the FIGO stage and residual tumor size. Lastly, parsimonious models, defined as simple models with great explanatory predictive power, are desirable, as are nomograms consisting of a minimal number of predictors with high predictive performance. As such, a parsimonious model may be useful in a clinical setting [32,33]; therefore, we regarded the six-variable models as the best models.

Interestingly, the primary treatment strategy was selected as one of the six variables related to platinum sensitivity. Specifically, NAC, rather than PDS, was associated with an increased risk of developing platinum resistance. Such a relationship is supported by the previous retrospective studies, which reported a higher rate of platinum-resistant recurrence in patients with stage IIIC-IV epithelial ovarian cancer who underwent NAC than those with PDS [34,35,36]. Although the underlying mechanism is not fully understood, researchers have suggested that NAC may increase ovarian cancer cell stemness and induce gene mutations towards drug resistance. This NAC-related platinum resistance may be further promoted by early exposure to chemotherapy when the tumor is still large or by remnant residual cancer cells, not completely resected at the time of interval debulking surgery [37].

In this study, four different machine learning methods, LR, RF, SVM, and DNN, were used. The performance of the LR-based model was comparable or superior to that of RF, SVM, or DNN. Similar findings were reported in other studies developing predictive models using patients’ clinicopathologic data [38,39], likely resulting from a weak signal-to-noise ratio associated with clinical prediction studies [40]. In a systematic review of 71 studies, Christodoulou et al. concluded that there is no evidence of the superior performance of RF, SVM, or DNN in clinical predictive models, relative to LR [40]. Further studies are warranted to justify the application of machine learning algorithms in developing clinical predictive models.

First-line treatments for HGSOC are changing. The addition of bevacizumab, a humanized anti-vascular endothelial growth factor monoclonal antibody, to the primary treatment of ovarian cancer improved the associated PFS [13,41]. Meanwhile, recent studies on PARP inhibitors have produced encouraging results. After complete or partial responses to platinum-based chemotherapy, maintenance with olaparib significantly decreased recurrence and mortality rates of patients with *BRCA1/2*-mutated advanced ovarian cancer [42]. Maintenance with niraparib has been associated with improved PFS, regardless of the *BRCA1/2* mutation status [14]. In addition, bevacizumab plus olaparib maintenance has shown PFS benefit in patients with homologous-recombination-deficient advanced ovarian cancer [15].

Regarding treatment options for newly diagnosed HGSOC, it remains unclear which maintenance therapy may be most suitable for which individual. Despite the survival benefit from both bevacizumab and PARP inhibitors, the toxicity and cost-effectiveness profiles of these therapies remain among their drawbacks. In addition, germline or somatic *BRCA1/2* mutation status may affect treatment efficacy; however, the required tests take time and other resources to perform. The present models may help physicians and patients in many aspects. In particular, the present models may enable the confirmation of platinum sensitivity before genetic test results are available. As the present study excluded patients treated with bevacizumab and PARP inhibitors, the proposed model is likely to predict the pure response to platinum-based chemotherapy. These predictions may facilitate the implementation of individualized treatment and surveillance protocols. The proposed models may also support clinical trial design.

The current study has several limitations. First, selection bias might exist due to the retrospective study design. Second, heterogeneity in patients and clinical practice among the tertiary hospitals is also problematic. For example, there were significant differences in the FIGO stage, proportion of patients who achieved complete cytoreduction, and OS among the institutions. In contrast, the distribution of the platinum-sensitive and -resistant patients was similar among the institutions. To overcome heterogeneity issues, we matched the institutions as well as the proportion of platinum-resistant cases at the time of data splitting for five-fold cross-validation. Third, molecular features, such as specific gene mutations, were not considered in this study. Regarding *BRCA1/2* mutation status, only a small portion of the patients underwent germline or somatic *BRCA1/2* gene testing in our institutions. By incorporating *BRRCA1/2* gene test results and other genetic alterations, we believe that it would be possible to develop predictive models with higher accuracy. Lastly, although we conducted internal validation using the five-fold cross-validation method, which is an established statistical approach, a further external validation study is warranted. Despite these limitations, we developed predictive models based on data from more than a thousand HGSOC patients with a relatively long observation period using various machine learning algorithms.

## 5. Conclusions

In conclusion, we developed machine learning models predicting platinum sensitivity in patients with HGSOC. Based on the six-variable LR model, we also developed a nomogram to facilitate clinical use of the proposed model. This nomogram is expected to support clinical practice, clinical trial design, and future research.

## Figures and Tables

**Figure 1 cancers-13-01875-f001:**
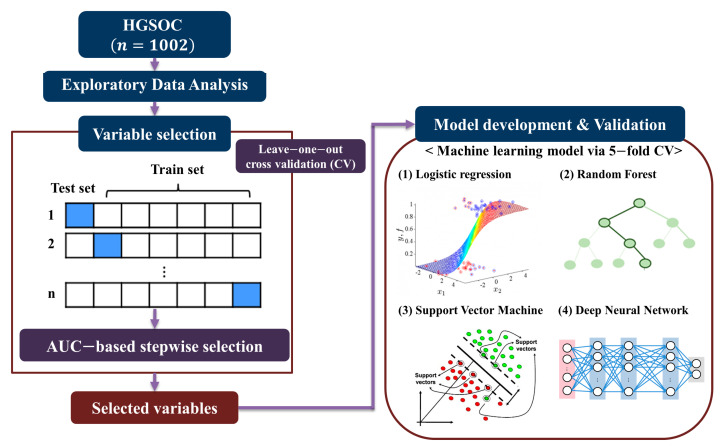
Overall workflow of statistical analysis.

**Figure 2 cancers-13-01875-f002:**
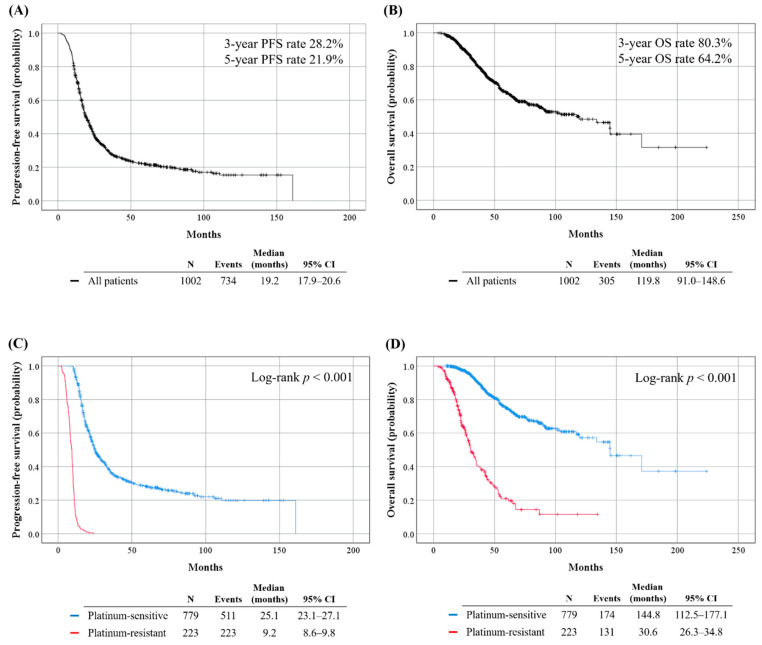
Survival outcomes of the study population. (Upper) All patients; (Lower) Comparisons between platinum-resistant and sensitive groups. (**A**,**C**) Progression-free survival; (**B**,**D**) Overall survival.

**Figure 3 cancers-13-01875-f003:**
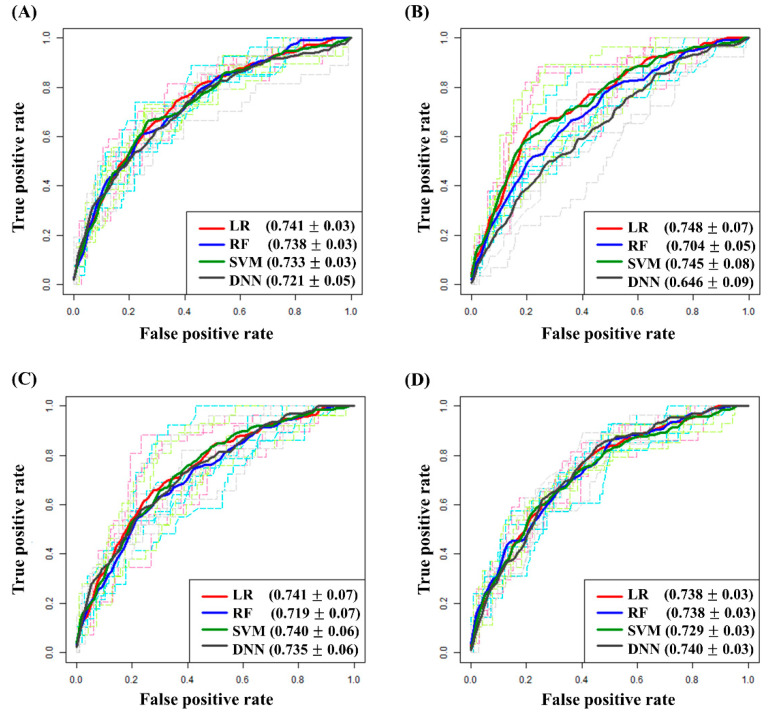
ROC curves for comparative models obtained via five-fold cross-validation. (**A**) Models consisting of six selected variables; (**B**) Models consisting of six selected variables and FIGO stage; (**C**) Models consisting of six selected variables and residual tumor size after debulking surgery; (**D**) Models consisting of six selected variables, FIGO stage, and residual tumor size after debulking surgery.

**Figure 4 cancers-13-01875-f004:**
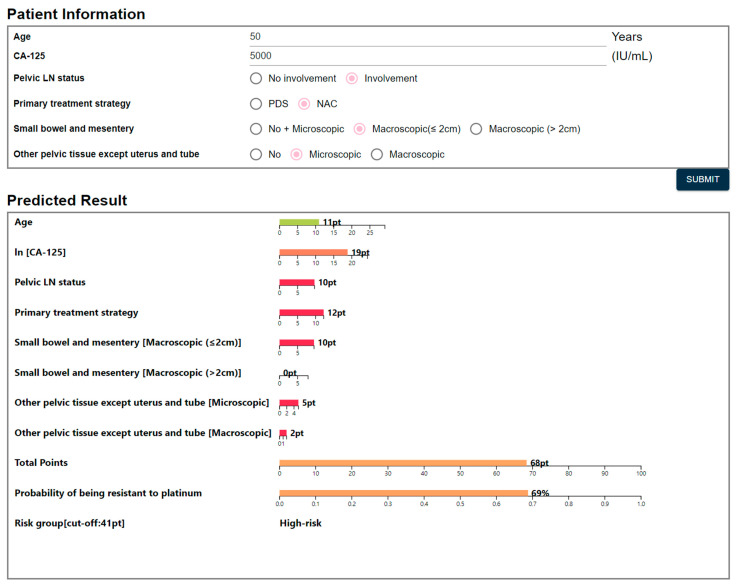
Web-based nomogram predicting platinum sensitivity.

**Table 1 cancers-13-01875-t001:** Patients’ clinicopathologic characteristics.

Characteristics	Missing Rate (%)	All (*n* = 1002, %)	Platinum-Sensitive (*n* = 779, %)	Platinum-Resistant (*n* = 223, %)	*p*
Age, years	0	55.8 ± 10	55.2 ± 10	58.3 ± 10	<0.001
BMI, kg/m^2^	3.2	23.6 ± 3	23.5 ± 3	23.9 ± 3	0.135
Parity	0.9				0.884
0		97 (9.8)	77 (10.0)	20 (9.1)	
1–2		592 (59.6)	458 (59.2)	134 (60.9)	
≥3		304 (30.6)	238 (30.8)	66 (30.0)	
Menopause	0.4	691 (69.2)	513 (66.1)	178 (80.2)	<0.001
Comorbidities					
Hypertension	22.2	153 (19.6)	107 (18.1)	46 (24.5)	0.069
Diabetes	22.2	53 (6.8)	36 (6.1)	17 (9.0)	0.215
Dyslipidemia	22.3	35 (4.5)	29 (4.9)	6 (3.2)	0.431
Personal history of breast cancer	3.3	72 (7.4)	59 (7.8)	13 (6.0)	0.453
Familial history of breast cancer *	5.5	51 (5.4)	42 (5.7)	9 (4.2)	0.486
No. of family members with cancer					
Median (range)	5.5	0 (0–3)	0 (0–3)	0 (0–2)	0.295
Familial history of gynecologic cancer *	5.5	21 (2.2)	18 (2.5)	3 (1.4)	0.511
No. of family members with cancer					
Median (range)	5.5	0 (0–2)	0 (0–2)	0 (0–2)	0.702
Origin	0				0.532
Ovary		911 (90.9)	708 (90.9)	203 (91.0)	
Tube		51 (5.1)	42 (5.4)	9 (4.0)	
Peritoneum		40 (4.0)	29 (3.7)	11 (4.9)	
FIGO stage	0				<0.001
I		40 (4.0)	40 (5.1)	0	
II		56 (5.6)	54 (6.9)	2 (0.9)	
III		628 (62.7)	496 (63.7)	132 (59.2)	
IV		278 (27.7)	189 (24.3)	89 (39.9)	
Ln (Serum CA-125 [IU/mL])	4.3	6.7 ± 2	6.6 ± 2	7.1 ± 1	<0.001
Hemoglobin (g/dL)	11.9	12.3 ± 1	12.3 ± 1	12.2 ± 1	0.192
Primary treatment strategy	0				<0.001
PDS		764 (76.2)	629 (80.7)	135 (60.5)	
NAC		238 (23.8)	150 (19.3)	88 (39.5)	
Residual tumor size after PDS/IDS	4.4				<0.001
Complete cytoreduction		549 (57.3)	455 (61.2)	94 (43.9)	
Gross residual tumor		409 (42.7)	289 (38.8)	120 (56.1)	
Frontline chemotherapy regimen	4.0				0.870
Paclitaxel-Carboplatin		872 (90.6)	676 (90.5)	196 (91.2)	
Docetaxel-Carboplatin		90 (9.4)	71 (9.5)	19 (8.8)	
Total cycle of frontline chemotherapy	0				<0.001
≤6		694 (69.3)	564 (72.4)	130 (58.3)	
>6		308 (30.7)	215 (27.6)	93 (41.7)	
Recurrence	0	734 (73.3)	511 (65.6)	223 (100.0)	<0.001
Treatment-free interval, months	0				
Median (range)		12.9 (0.1–153.4)	17.2 (6.1–153.4)	3.4 (0.1–6.0)	<0.001

Data are presented as mean ± standard deviation for continuous variables and as count (%) for categorical variables. Abbreviations: BMI, body mass index; CA-125, cancer antigen 125; FIGO, International Federation of Gynecology and Obstetrics; IDS, interval debulking surgery; NAC, neoadjuvant chemotherapy; PDS, primary debulking surgery. ***** Up to second degree.

**Table 2 cancers-13-01875-t002:** Performance of developed models identifying platinum-resistant cases.

No. of Variables	List	Machine Learning	AUC	Sensitivity	Specificity	Balanced Accuracy	Threshold
1	FIGO stage	LR	0.556	1.000	0.111	0.556	0.025
		RF	0.500	0	1.000	0.500	0
		SVM	0.556	1.000	0.167	0.583	0.241
		DNN	0.558	1.000	0.122	0.561	0.214
1	Residual tumor size after PDS/IDS	LR	0.586	0.605	0.564	0.584	0.172
		RF	0.500	0	1.000	0.500	0
		SVM	0.586	0.605	0.564	0.584	0.295
		DNN	0.587	0.605	0.564	0.584	0.355
2	FIGO stage + Residual tumor size after PDS/IDS	LR	0.611	0.605	0.570	0.588	0.203
		RF	0.500	0	1.000	0.500	0
		SVM	0.611	0.605	0.570	0.588	0.309
		DNN	0.611	0.605	0.570	0.588	0.252
6	Age + Serum CA125 levels * + NAC + Pelvic LN status + Involvement of pelvic tissue other than uterus and tube + Involvement of small bowel and mesentery	LR	0.741	0.778	0.622	0.700	0.175
		RF	0.738	0.538	0.887	0.713	0.185
		SVM	0.733	0.731	0.745	0.738	0.232
		DNN	0.721	0.857	0.556	0.706	0.357
7	Age + Serum CA125 levels * + NAC + Pelvic LN status + Involvement of pelvic tissue other than uterus and tube + Involvement of small bowel and mesentery + FIGO stage	LR	0.748	0.920	0.476	0.698	0.141
		RF	0.704	0.800	0.524	0.662	0.034
		SVM	0.745	0.920	0.457	0.689	0.133
		DNN	0.646	0.655	0.625	0.640	0.218
7	Age + Serum CA125 levels * + NAC + Pelvic LN status + Involvement of pelvic tissue other than uterus and tube + Involvement of small bowel and mesentery + Residual tumor size after PDS/IDS	LR	0.741	0.793	0.563	0.678	0.144
		RF	0.719	0.960	0.385	0.672	0.021
		SVM	0.740	0.517	0.883	0.700	0.259
		DNN	0.735	0.680	0.654	0.667	0.461
8	Age + Serum CA125 levels * + NAC + Pelvic LN status + Involvement of pelvic tissue other than uterus and tube + Involvement of small bowel and mesentery + FIGO stage + Residual tumor size after PDS/IDS	LR	0.738	0.769	0.648	0.708	0.211
		RF	0.738	0.897	0.519	0.708	0.065
		SVM	0.729	0.519	0.883	0.701	0.293
		DNN	0.740	0.852	0.561	0.707	0.088

Abbreviations: AUC, area under the receiver operating characteristic curve; CA-125, cancer antigen 125; DNN, deep neural network; FIGO, International Federation of Gynecology and Obstetrics; IDS, interval debulking surgery; LN, lymph node; LR, logistic regression; NAC, neoadjuvant chemotherapy; PDS, primary debulking surgery; RF, random forest; SVM, support vector machine. ***** Ln-transformed.

## Data Availability

The data presented in this study are available on request from the corresponding authors.

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
