# Peer review of "Development of Machine Learning Models to Predict Platinum Sensitivity of High-Grade Serous Ovarian Carcinoma"

_cancers, 2021, doi:10.3390/cancers13081875_

Round 1

Reviewer 1 Report

Hwangbo et al. describe in their manuscript the development of a prediction model and nomogram for platinum resistance of high-grade serous ovarian cancer using clinicopathologic data. The authors used 6 out of 42 analyzed parameters from 1002 patients to establish this model. Platinum resistance is one of the major causes for HGSOC mortality and more research is needed to increase the understanding of resistance development and improve the prognosis of patients with resistant disease. However, this reviewer sees major limitations of the study. 

  1. The developed model includes the parameter of primary treatment strategy (neoadjuvant chemotherapy (NACT) or primary surgery (PS)). However, the specific indication for neoadjuvant chemotherapy is controversially discussed. Thus, the primary treatment strategy is not an objectively measured/collected parameter. The authors should try to establish specific models for patients after NACT or PS.
  2. The study population seems to be not comparable to HGSOC patients in general. The median OS of 120 month is rather high for HGSOC and the 5-year survival rate of 64.2% is similar to the rate for all epithelial ovarian cancer patients in South Korea. The authors should discuss reasons for the exceptional good prognosis of this cohort and potential limitations of their model.
  3. Albeit the authors applied internal cross validations an external validation would strengthen the confidence in their model- specifically for the use in other populations. They should validate their model using independent datasets (some are mentioned in the manuscript or Wahner Hendrickson et al. Gynecol Oncol 2015). If no other datasets are available they may validate their model using data from their publication (Kim SI et al. Cancer Res. Treat. 2019) if the HGSOC patients (n=401) are independent from the present cohort.
  4. The authors state sensitivity and specificity for the identification of resistant patients in Table 2. Which cut-off for the nomogram-based risk score was applied? This cut-off and the associated sensitivity and specificity values should be stated within the nomogram.
  5. Aim of all predictive or prognostic models should be an improved outcome for patients after individualized/adapted treatment strategies. Thus, models should clearly discriminate between patients with good and worse outcome after standard therapy. Applying the cut-off (see point 4) the authors should compare PFS and OS for the identified subgroups. Moreover, these results should be discussed in comparison to other (prognostic) models from literature applied to their own data.

Minor remarks

  1. Published data suggest that NACT itself can induce resistance or increase the risk of developing platinum resistance. This should be discussed.
  2. Within the material and method section (line 106) the authors state about the primary treatment. Information should be added that 238 patients were treated with NACT.
  3. To enable a better evaluation of their model, the authors may include DCA plots in their manuscript.

Author Response

Please see the attachment for details.

Reviewer 1
Hwangbo et al. describe in their manuscript the development of a prediction model and nomogram for platinum resistance of high-grade serous ovarian cancer using clinicopathologic data. The authors used 6 out of 42 analyzed parameters from 1002 patients to establish this model. Platinum resistance is one of the major causes for HGSOC mortality and more research is needed to increase the understanding of resistance development and improve the prognosis of patients with resistant disease. However, this reviewer sees major limitations of the study.
Comment 1: The developed model includes the parameter of primary treatment strategy (neoadjuvant chemotherapy (NACT) or primary surgery (PS)). However, the specific indication for neoadjuvant chemotherapy is controversially discussed. Thus, the primary treatment strategy is not an objectively measured/collected parameter. The authors should try to establish specific models for patients after NACT or PS.
Answer 1:
We deeply appreciate your thorough review and valuable comments. We totally agree with your opinion that the primary treatment strategy is not an objectively measured/collected parameter. However, since the publication of two monumental randomized clinical trials, EORTC [1] and CHORUS [2], NAC followed by interval debulking surgery has been considered as an acceptable standard of care for women with advanced-stage epithelial ovarian cancer who are not suitable for PDS. Despite controversies surrounding NAC, it is true that proportion of NAC gradually increased in real-world clinical practice. According to a time-series analysis study including more than 72,000 women treated for advanced epithelial ovarian cancer, the use of NAC increased from 17.6% in 2004 to 45.1% in 2016 [3]. In the current study, a total of 1,002 patients were included from the three tertiary institutional hospitals in Korea. The three institutions are conducting NAC on a similar basis. In general, we apply NAC when the patients meet one of the following conditions: 1) high tumor burden on initial imaging studies with multiple and unresectable extra-abdominal metastases, multiple liver parenchymal metastases or pulmonary metastases, and extensive small bowel/mesenteric root involvement; 2) poor performance status and high operative risk with severe comorbidities, or 3) not suitable for PDS, where expected suboptimal cytoreduction (residual disease measuring >1 cm). As the results, 23.8% of the study population received NAC followed by interval debulking surgery (n=238). Like other variables, primary treatment strategy (NAC vs. PDS) underwent the variable selection processes, and was selected as one of the six variables related to platinum sensitivity. Previously, our research team developed nomograms predicting platinum sensitivity, 3-year PFS, and 5-year OS in patients with epithelial ovarian cancer. Consistent with the current study, In that study, NAC was not only incorporated in nomogram for platinum sensitivity, but also in nomograms for 3-year PFS and 5-year OS [4].
In our humble opinion, we consider NAC as a variable somewhat representing or reflecting real-world clinical practice. Models developed in NAC only populations or PDS only population are insufficient models, which does not accurately reflect the real-world clinical practice. Moreover, a unified model is more convenient for clinical utility, compared to the separate models. Lastly, to apply various machine learning methods, sample size should be large enough: however, if we split the study population into NAC and PDS subsets, reduction of the sample size is inevitable, so that it can be disadvantageous in terms of predictive power. Please understand our situations.
As per the reviewer’s comments, we added the following sentence in Methods section:
(Page 5, Lines 214-216) “In terms of primary treatment, 764 (76.2%) patients underwent PDS, while 238 (23.8%) patients received NAC followed by interval debulking surgery”.
We also added the following sentence in Results section: (Page 3, Lines 125-131) “The three institutions are conducting NAC on a similar basis. In general, we apply NAC when the patients meet one of the following conditions: 1) high tumor burden on initial imaging studies with multiple and unresectable extra-abdominal metastases, multiple liver parenchymal metastases or pulmonary metastases, and extensive small bowel/mesenteric root involvement; 2) poor performance status and high operative risk with severe comorbidities, or 3) not suitable for PDS, where expected suboptimal cytoreduction (residual disease measuring >1 cm)”.

Comment 2:
The study population seems to be not comparable to HGSOC patients in general. The median OS of 120 month is rather high for HGSOC and the 5-year survival rate of 64.2% is similar to the rate for all epithelial ovarian cancer patients in South Korea. The authors should discuss reasons for the exceptional good prognosis of this cohort and potential limitations of their model.
Answer 2:
We appreciate your valuable comments.
The aim of this study was to develop machine learning models predicting platinum sensitivity in patients with HGSOC. Actually, patients’ OS was not considered in this study, until your comments. Thank you once again for your guidance.
For the main purpose of our study, we excluded the patients who did not receive platinum-based chemotherapy. To determine platinum sensitivity, we also excluded those who were lost to follow-up before reaching 6 months of platinum-free interval without evidence of disease recurrence. In other words, it means that selection bias definitely exists in this study, which might affect prolonged median OS. As you pointed out, the 5-year survival rate of the current study was similar to those observed in the Korea Central Cancer Registry. According to the Korea Central Cancer Registry, the 5-year relative survival rate of ovarian cancer for patients diagnosed with ovarian cancer in the recent 5 years, from 2013 to 2017 was 64.9% in Korea [5].
We also recognize that institutional heterogeneity might exist and affect survival outcomes. All three institutions in this study are so called “high volume centers”, representing Korea. However, as shown in the survival curves below, significant difference in OS was observed among the three institutions: patients who were treated at the institution A showed much enlogated OS, compared to other institutions.

At the same time, we also observed significant differences in FIGO stage and residual tumor size after cytoreductive surgery among the three institutions, which suggest presence of severe institutional heterogeneity.

We admit such heterogeneity in the study population is definitely one of the limitations of the current study. Neverthless, we did not mention it in the original manuscript, we considered the proportion of platinum responses and the proportion of institutions when splitting the dataset for 5-fold cross-validation to reduce the heterogeneity.
As per the reviewer’s comments, we added the following sentences in Methods and Discussion sections to clarify this issue. (Page 4, Lines 194-195) “Splitting the whole dataset for 5-fold cross-validation, we considered the proportions of platinum-resistant cases and institutions to reduce the heterogeneity”. (Page12, Lines 389-396) “Second, heterogeneity in patients and clinical practice among the tertiary hospitals is also problematic. For example, there were significant differences in FIGO stage, proportion of the patients who achieved complete cytoreduction, and OS among the institutions. In contrast, the distribution of the platinum-sensitive and -resistant patients was similar among the institutions. To overcome heterogeneity issues, we matched the institutions as well as the proportion of platinum-resistant cases at the time of data splitting for 5-fold cross-validation”.

Comment 3: Albeit the authors applied internal cross validations, an external validation would strengthen the confidence in their model- specifically for the use in other populations. They should validate their model using independent datasets (some are mentioned in the manuscript or Wahner Hendrickson et al. Gynecol Oncol 2015). If no other datasets are available they may validate their model using data from their publication (Kim SI et al. Cancer Res. Treat. 2019) if the HGSOC patients (n=401) are independent from the present cohort.
Answer 3: We appreciate your guidance. We totally agree with your opinion that external validation in independent datasets is needed.
Previously, our research team developed a nomogram predicting platinum sensitivity using clinicopathologic data of 710 patients with epithelial ovarian cancer who were treated at two tertiary institutional hospitals (SNUH and AMC), including 389 (54.8%) with HGSOC [4].
Among the current study population (n=1002), 388 patients from SNUH and AMC were also part of the study population assessed in our previous study [4]. We re-reviewed these patients’ clinicopathologic and survival data for the current study. The other 614 patients were newly included from the three tertiary institutional hospitals, SNUH, AMC, and Severance. Due to the overlapping of patients, it was impossible to conduct an external validation of the developed models in the study population of our previous study. Meanwhile, we could not secure other datasets for external validation, either, during the study and revision periods. We only could conduct the AUC-based stepwise selection method in variable selection, and based on these variables, models predicting platinum sensitivity was developed via five-fold cross-validation
Please understand our situations. We will conduct an external validation study of our findings as soon as an independent dataset is available.
As per the reviewer’s comments, we clarified this issue in Methods and Discussion section of the revised manuscript.
(Page 5, Lines 212-213) “Among the study population, 388 (38.7%) were also part of the study population assessed in our previous study.
(Page 12, Lines 410-403) “Lastly, although we conducted internal validation using the 5-fold cross-validation method, which is an established statistical approach, a further external validation study is warranted”.

Comment 4: The authors state sensitivity and specificity for the identification of resistant patients in Table 2. Which cut-off for the nomogram-based risk score was applied? This cut-off and the associated sensitivity and specificity values should be stated within the nomogram.
Answer 4: We thank the reviewer for pointing out this issue. As mentioned in Materials and Methods, we chose a threshold with the maximum balanced accuracy, defined as the average of sensitivity and specificity (Page, Lines). The cut-off value of the nomogram or LR-based model was 0.175, and the corresponding sensitivity, specificity, and balanced accuracy were 0.778, 0.622, and 0.700, respectively. In response to the reviewer’s comment, we added the corresponding cutoff (i.e. threshold) for each model in Table 2.

Comment 5: Aim of all predictive or prognostic models should be an improved outcome for patients after individualized/adapted treatment strategies. Thus, models should clearly discriminate between patients with good and worse outcome after standard therapy. Applying the cut-off (see point 4) the authors should compare PFS and OS for the identified subgroups. Moreover, these results should be discussed in comparison to other (prognostic) models from literature applied to their own data.
Answer 5: We appreciate your valuable comments and guidance. As per the reviewer’s comments, we compared PFS and OS between the platinum-sensitive and -resistant groups predicted by the six-variable, LR model, which showed the best performance in identifying platinum-resistant cases in this study. Herein, the cut-off value of 0.175 was used. The Figures below present comparisons of PFS and OS according to the predicted platinum-sensitive and -resistant groups. As we conducted five-fold cross-validation, five sets of survival curves were generated. Consequently, significant differences in PFS and OS were observed between the two groups. These results demonstrate that the developed predictive model also discriminates patients with good and poor survival outcomes well.
We added these points to Results section of the revised manuscript as follows: (Page 9, Lines 280-285) “Next, we compared patients’ survival outcomes between platinum-sensitive and -resistant groups predicted by the six-variable, LR model with the cut-off value of 0.175 (Figures S2). Regardless of the validation set, the platinum-resistant group showed significantly worse PFS and OS than the platinum-sensitive group. These results suggest that the developed predictive model also discriminates patients with good and poor survival outcomes well”.
We also compared our six-variable, LR model with other predictive models in the literature using the current study population consisting of only HGSOC. To date, including one published from our research team, only two models have been developed to predict platinum sensitivity in epithelial ovarian cancer [4, 6]. Both nomograms from Paik et al.’s and Kim et al’s studies used all histologic types of epithelial ovarian cancer, rather than a specific subtype. While Paik et al’s model targeted only PDS cases, Kim et al’s included both PDS and NAC cases. Moreover, the selected variables were also different among the studies, which further makes it difficult to conduct fair comparisons.

Thus, we conducted the following procedures, independently. (1) Complete dataset: We used the complete set after excluding samples with missing values (2) Imputed dataset: We imputed missing variables not available in the current study population. For imputation, we used multivariate imputation by the chained equations (MICE) algorithm. This algorithm is one of the commonly used method and imputes data on a variable-by-variable basis by specifying an imputation model per variable. We generated five imputed datasets via the MICE algorithm and calculated the average of AUCs for five imputed datasets. Applying the previous model developed by our research team, we used the same beta coefficients. As the results, the previous model from our research team showed a performance identifying platinum-resistant cases with AUCs of 0.667 and 0.639 in the complete dataset and the imputed dataset, respectively. Lastly, we observed that the LR-based model of the current study showed significantly better predictive performance than the previous model (AUCs, 0.762 vs. 0.667 in complete dataset; 0.706 vs. 0.639 in imputed dataset.
(A) Complete dataset (B) Imputed dataset In comparisons between our six-variable, LR model and Paik et al’s model, the proposed model showed higher AUCs than those of Paik et al’s model, both in the complete dataset and the imputed dataset, respectively. However, statistical significance was not observed.
(A) Complete dataset (B) Imputed dataset
We added the following sentences to Discussion section as follows: (Page 10, Lines 307-319) “Previous studies aimed at developing predictive models of platinum sensitivity did not confine ovarian cancer patients to a specific histologic type, such as HGSOC [4, 6]. While Paik et al.’s study included only PDS cases [6], our previous and current studies put PDS and NAC cases together in the model development [4]. The selected variables also differed among the models [4, 6]. Such differences in study population and selected variables make it difficult to conduct comparisons of predictive models even in the same population. Nevertheless, we could compare the predictive performance of models in the current study population by excluding samples with missing values (complete dataset) or generating imputed datasets using multivariate imputation by the chained equations algorithm. As the results, the six-variable, LR model of the current study had significantly superior predictive power than our previous model (AUCs, 0.762 vs. 0.667; p = 0.009 in the complete dataset; 0.706 vs. 0.639; p <0.05 in the imputed dataset), but similar to Paik et al.’s model”.

Minor remarks
Comment 6:
Published data suggest that NACT itself can induce resistance or increase the risk of developing platinum resistance. This should be discussed.
Answer 6:
Thank you for your guidance.
As per the reviewer’s comments, we added the following sentences in Discussion section:
(Page 11, Lines 347-356) “Interestingly, primary treatment strategy was selected as one of the six variables related to platinum sensitivity. Specifically, NAC, rather than PDS, was associated with increased risk of developing platinum resistance. Such relationship is supported by the previous retrospective studies, which reported a higher rate of platinum-resistant recurrence in patients with stage IIIC-IV epithelial ovarian cancer who underwent NAC than those with PDS [7-10]. Although the underlying mechanism is not fully understood, researchers have suggested that NAC may increase ovarian cancer cell stemness and induce gene mutations towards drug resistance. This NAC-related platinum resistance may be further promoted by early exposure to chemotherapy when the tumor is still large or by remnant residual cancer cells, not completed resected at the time of interval debulking surgery [11]”.

Comment 7:
Within the material and method section (line 106) the authors state about the primary treatment. Information should be added that 238 patients were treated with NACT.
Answer 7:
We appreciate your guidance.
We reworded one of the inclusion criteria in Methods section as follows:
(Page 3, Lines 109-112) “(4) underwent primary treatment, either primary debulking surgery (PDS) followed by platinum-based postoperative adjuvant chemotherapy or platinum-based neoadjuvant chemotherapy (NAC) followed by interval debulking surgery and postoperative adjuvant chemotherapy”.
our your guidance
We also added the following sentence in Results section as follows:
(Page 5, Lines 214-215) “In terms of primary treatment, 764 (76.2%) patients underwent PDS, while 238 (23.8%) patients received NAC followed by interval debulking surgery”.

Comment 8:
To enable a better evaluation of their model, the authors may include DCA plots in their manuscript.
Answer 8: As per the reviewer’s comments, we presented the multidimensional scaling plot, which is similar to DCA plot but more popularly used to display patterns in multivariate data. Here are the MDS plots: (A) For 42 independent variables that underwent exploratory analysis; and (B) For six selected variables in this study. Unfortunately, it was difficult to separate the platinum-sensitive and -resistant groups through these MDS plots.

Thank you for your consideration. We look forward to hearing from you.

(Ref.)

[1] Vergote I, Tropé CG, Amant F, Kristensen GB, Ehlen T, Johnson N, et al. Neoadjuvant chemotherapy or primary surgery in stage age IIIC or IV ovarian cancer. The New England journal of medicine. 2010;363:943IIIC or IV ovarian cancer. The New England journal of medicine. 2010;363:943--53.53.
[2] Kehoe S, Hook J, Nankivell M, Jayson GC, Kitchener H, Lopes T, et al. Primary chemotherapy versus primary surgery for newly ll M, Jayson GC, Kitchener H, Lopes T, et al. Primary chemotherapy versus primary surgery for newly diagnosed advanced ovarian cancer (CHORUS): an opendiagnosed advanced ovarian cancer (CHORUS): an open--label, randomised, controlled, nonlabel, randomised, controlled, non--inferiority trial. Lancet (London, England). inferiority trial. Lancet (London, England). 2015;386:2492015;386:249--57.57.
[3] Melamed A, Rauhamed A, Rauh--Hain JA, Knisely AT, St. Clair CM, Tergas AI, Collado FK, et al. The effect of liberal versus restrictive use of Hain JA, Knisely AT, St. Clair CM, Tergas AI, Collado FK, et al. The effect of liberal versus restrictive use of neoadjuvant chemotherapy (NACT) for ovarian cancer on postoperative mortality and longneoadjuvant chemotherapy (NACT) for ovarian cancer on postoperative mortality and long--term survival: A quasiterm survival: A quasi--experimental study. experimental study. Gynecologic oncology. 2020;159:81.Gynecologic oncology. 2020;159:81.
[4] Kim SI, Song M, Hwangbo S, Lee S, Cho U, Kim JH, et al. Development of Web--Based Nomograms to Predict Treatment Response Based Nomograms to Predict Treatment Response and Prognosis of Epithelial Ovarian Cancer. Cancer research and treatment. 2019;51:1144and Prognosis of Epithelial Ovarian Cancer. Cancer research and treatment. 2019;51:1144--55.55.
[5] Hong S, Won YJ, Park YR, Jung KW, Kong HJ, Lee ES. Cancer Statistics in Korea: Incidence, Mortality, Survival, and PrevalenceHong S, Won YJ, Park YR, Jung KW, Kong HJ, Lee ES. Cancer Statistics in Korea: Incidence, Mortality, Survival, and Prevalence in 2017. Cancer research and treatment. 2020;52:335in 2017. Cancer research and treatment. 2020;52:335--50.50.
[6] Paik ES, Sohn I, Baek SY, Shim M, Choi HJ, Kim TJ, et al. Nomograms Predicting Platinum Sensitivity, Progressionedicting Platinum Sensitivity, Progression--Free Survival, Free Survival, and Overall Survival Using Pretreatment Complete Blood Cell Counts in Epithelial Ovarian Cancer. Cancer research and treatmenand Overall Survival Using Pretreatment Complete Blood Cell Counts in Epithelial Ovarian Cancer. Cancer research and treatment. t. 2017;49:6352017;49:635--42.42.
[7] Petrillo M, Ferrandina G, Fagotti A, Vizzielli G, Margariti PA, Pedone AL, et al. Timing and pattern of recurrence in ovarian cancer Margariti PA, Pedone AL, et al. Timing and pattern of recurrence in ovarian cancer patients with high tumor dissemination treated with primary debulking surgery versus neoadjuvant chemotherapy. Annals of surgpatients with high tumor dissemination treated with primary debulking surgery versus neoadjuvant chemotherapy. Annals of surgical ical oncology. 2013;20:3955oncology. 2013;20:3955--60.60.
[8] da Costa AA, Valadares CV, Baiocchi G, Mantoan H, Saito A, Sanches S, et al. Neoadjuvant Chemotherapy Followed by Interval Valadares CV, Baiocchi G, Mantoan H, Saito A, Sanches S, et al. Neoadjuvant Chemotherapy Followed by Interval Debulking Surgery and the Risk of Platinum Resistance in Epithelial Ovarian Cancer. Annals of surgical oncology. 2015;22 SuppDebulking Surgery and the Risk of Platinum Resistance in Epithelial Ovarian Cancer. Annals of surgical oncology. 2015;22 Suppl l 3:S9713:S971--8.8.
[9] Luo Y, Lee M, Kim HS, Chung HH, Song YS. Effect of neoadjuvant chemotherapy on platinum resistance in stage IIIC and IV Y, Lee M, Kim HS, Chung HH, Song YS. Effect of neoadjuvant chemotherapy on platinum resistance in stage IIIC and IV epithelial ovarian cancer. Medicine. 2016;95:e4797.epithelial ovarian cancer. Medicine. 2016;95:e4797.
[10] Gao Y, Li Y, Zhang C, Han J, Liang H, Zhang K, et al. Evaluating the benefits of neoadjuvant chemotherapy for advanced epithelial neoadjuvant chemotherapy for advanced epithelial ovarian cancer: a retrospective study. Journal of ovarian research. 2019;12:85.ovarian cancer: a retrospective study. Journal of ovarian research. 2019;12:85.
[11] Liu J, Jiao X, Gao Q. Neoadjuvant chemotherapy--related platinum resistance in ovarian cancer. Drug discovery today. related platinum resistance in ovarian cancer. Drug discovery today. 2020;22020;25:12325:1232--8.8.

Reviewer 2 Report

The authors have presented a well-organized approach to predict platinum sensitivity in HGSOC patients using ML. The overall presentation, quality of figures and readability is excellent, and will be understandable to readers with distinct backgrounds. Without undermining the impact of the work presented, my biggest concern is with the novelty of the work - both in terms of findings as well as the ideas. There are places where the authors can highlight how the current work builds upon and improves on the previous ML models predicting platinum sensitivity. Other than that, there a few minor comments I would like the authors to address: 1. Lines 92 - The authors don't make it clear why its "necessary to focus on the HGSOC" in context of their previous study where the authors had already trained a model to predict platinum sensitivity. Was the mode performance not sufficient, or is there a strong biological distinction that causes the model to differ between serous and HGSOC subtypes? 2. Line 114 - Are these the same as those listed in Tables S1-2? If yes, the authors should make a reference to the supplementary tables in line with the statement. 3. Line 193 - Distribution of patients across sources could be shown as box plots in supplementary data 4. In line with the above statement, it might be relevant to check if any of the 42 input variables have skews across the 3 institutional sources. 5. Line 143 (Variable Selection Method) - Is this iterative methodology of variable selection attributable to an earlier study, if so, the authors should reference it. If it has been developed particularly for this purpose, the authors should explain as to why this method is an improvement over the currently acceptable methods of recursive feature selection. 6. Line 240 - The authors do not offer an explanation as to why FIGO stage and residual tumor size weren't selected as model inputs by the variable selection method. Objectively, should this mean that the prediction of sensitivity is invariant to these parameters. In fact, this is one place where authors should highlight that popular prognostic markers such as these did NOT affect the prediction accuracy of the ML models. 7. Finally, did the authors not consider including any molecular features for prediction. Given that not all patients are tested for mutations etc, for the patients that do have the available information, would that affect the outcome for patients? This in fact could guide the inclusion of looking at molecular markers in deciding treatment options.

Author Response

Reviewer 2
The authors have presented a well-organized approach to predict platinum sensitivity in HGSOC patients using ML. The overall presentation, quality of figures and readability is excellent, and will be understandable to readers with distinct backgrounds. Without undermining the impact of the work presented, my biggest concern is with the novelty of the work - both in terms of findings as well as the ideas. There are places where the authors can highlight how the current work builds upon and improves on the previous ML models predicting platinum sensitivity. Other than that, there a few minor comments I would like the authors to address:

Comment 1: Lines 92 - The authors don't make it clear why its "necessary to focus on the HGSOC" in context of their previous study where the authors had already trained a model to predict platinum sensitivity. Was the mode performance not sufficient, or is there a strong biological distinction that causes the model to differ between serous and HGSOC subtypes?

Answer 1:
Thank you for your thorough review and words of encouragement.
Epithelial ovarian cancer is a histologically very heterogeneous disease with distinct features. Among the various types, high-grade serous ovarian carcinoma (HGSOC) is the most common and aggressive histological subtype [1]. Currently, most of the omics studies have focused on HGSOC [2, 3]. Therefore, we considered that development of models with high predictive power focusing on HGSOC is very necessary in accordance with the era of precision cancer medicine.
We evaluated predictive performance of the previous model for platinum sensitivity [4], which was developed by our research team using clinicopathologic data of 710 epithelial ovarian cancer patients with various histologic types, in the current study population consisting of only HGSOC. The same beta coefficients from the previous model were applied. We also compared predictive performance of the previous model and the LR-based model of the current study in the same population. As some variables in the previous models were quite missing in the current study population (e.g., ascites/peritoneal washing cytology), we conducted the following procedures, independently. (1) Complete dataset: We used the complete set after excluding samples with missing values (2) Imputed dataset: We imputed missing variables not available in the current study population. For imputation, we used multivariate imputation by the chained equations (MICE) algorithm. This algorithm is one of the commonly used method and imputes data on a variable-by-variable basis by specifying an imputation model per variable. We generated five imputed datasets via the MICE algorithm and calculated the average of AUCs for five imputed datasets. As the results, the previous model showed a performance identifying platinum-resistant cases with AUCs of 0.667 and 0.639 in the complete dataset and the imputed dataset, respectively. Lastly, we observed that the LR-based model of the current study showed significantly better predictive performance than the previous model (AUCs, 0.762 vs. 0.667 in complete dataset; 0.706 vs. 0.639 in imputed dataset.

(A) Complete dataset (B) Imputed dataset
As per the reviewer’s comments, we enhanced Introduction section by rewording and adding sentences to clarify the necessity of developing predictive models in HGSOC as follows:
(Page 2, Lines 88-98) “Previously, our research team developed a nomogram predicting platinum sensitivity using clinicopathologic data of 710 patients with epithelial ovarian cancer, including 389 (54.8%) with HGSOC [4]. In that study, the model showed a performance identifying platinum-resistant cases with the area under the receiver operating characteristic (ROC) curve (AUC) of 0.758. However, applying this model to an extended cohort, we observed a significant drop in the predictive performance. Considering that most multi-omics studies on ovarian cancer have been conducted in HGSOC, it is necessary to focus on the HGSOC and develop models predicting platinum sensitivity in this patient group”.

Comment 2: Line 114 - Are these the same as those listed in Tables S1-2? If yes, the authors should make a reference to the supplementary tables in line with the statement.

Answer 2:
Thank you for your guidance.
We extracted clinicopathologic data from patients’ medical records and pathologic reports. Detailed information on the type of collected variables was the same as our previous study [4] except for differential blood cell counts at initial diagnosis, which were not collected in this study.
As per the reviewer’s comments we added the statement with a reference to the Table S1 and S2.

Comment 3: Line 193 - Distribution of patients across sources could be shown as box plots in supplementary data.
Answer 3: We appreciate your guidance. The distribution of the platinum-sensitive and -resistant patients among institutions is shown in the table below.

As per the reviewer’s comments, we prepared bar plots as Figure S1.

Comment 4: In line with the above statement, it might be relevant to check if any of the 42 input variables have skews across the 3 institutional sources.
Answer 4: We thank the reviewer for pointing out this issue. In response to the reviewer’s comment, we checked for differences in the 42 input variables among three institutions. The following figure is a multidimensional scaling (MDS) plot. As shown this figure, we could observe that the data were evenly distributed across the institutions.

Comment 5: Line 143 (Variable Selection Method) - Is this iterative methodology of variable selection attributable to an earlier study, if so, the authors should reference it. If it has been developed particularly for this purpose, the authors should explain as to why this method is an improvement over the currently acceptable methods of recursive feature selection.
Answer 5: We thank the reviewer for pointing out this issue. The AUC-based stepwise variable selection method was also used in our previously published studies [4, 5]. As per the reviewer’s comments, we added the references to the revised manuscript.

Comment 6: Line 240 - The authors do not offer an explanation as to why FIGO stage and residual tumor size weren't selected as model inputs by the variable selection method. Objectively, should this mean that the prediction of sensitivity is invariant to these parameters. In fact, this is one place where authors should highlight that popular prognostic markers such as these did NOT affect the prediction accuracy of the ML models.
Answer 6: We appreciate your valuable comments. We recognize that FIGO stage and residual tumor size are important, well-known prognostic factors for survival outcomes in ovarian cancer.
First, please see Table 2. As shown here, LR-based predictive model consisting of two variables, FIGO stage and residual tumor size, showed a performance identifying platinum-resistant cases with AUC of 0.611. In this study, we conducted the variable selection processes. At this step, all 42 variables were evaluated fairly without giving any priority to the two variables, FIGO stage and residual tumor size. As the results, the following six variables were selected: age (continuous), serum CA-125 levels (continuous), primary treatment strategy (NAC vs. PDS), pelvic LN status (metastasis vs. no metastasis), tumor involvement of pelvic tissue other than uterus and tube (macroscopic vs. microscopic or no involvement), and tumor involvement of the small bowel and mesentery (>2 cm vs. ≤2 cm). Then, we developed LR-based predictive model using these six variables. This model showed better performance identifying platinum-resistant cases with AUC of 0.741.
We further added FIGO stage and residual tumor size after debulking surgery, and developed LR-based, eight-variable predictive models. However, the eight-variable model showed poor performance than the six-variable model (AUC, 0.738 vs. 0.741). These results suggest that FIGO stage and residual tumor size were not chosen as they did not contribute to further improvement of predictive performance.
Table 2. Performance of developed models identifying platinum-resistant cases.

Next, please see each selected variables. The variable, named “pelvic LN status” is directly associated with FIGO stage, as patients with pelvic LN metastasis is assigned to at least stage IIIA. The variable, named “involvement of pelvic tissue other than uterus and tube” is also directly associated with FIGO stage, as patients with this is assigned to al least stage IIA. The variable, “tumor involvement of the small bowel and mesentery” is also directly associated with FIGO stage, as patients with this is assigned to at least stage III. Moreover, >2cm tumor of the small bowel and mesentery is associated with residual tumor as it is difficult to remove them completely.
Putting together, the combination of the six selected variables seems to be more suitable for reflecting disease status and consequently shows better predictive performance than the popular prognostic markers.
As per the reviewer’s comments, we added the following sentences in Discussion section: (Page 11, Lines 325-328) “During the AUC-based stepwise selection, where all 42 independent variables were evaluated fairly without giving any priority to the specific variables, the six variables were selected. In contrast, both FIGO stage and residual tumor size after debulking surgery, which are the best known prognostic factors in ovarian cancer, were not selected”. (Page 11, Lines 335-342) “At the same time, we cannot help but point out that among the six selected variables, “pelvic LN status”, “involvement of pelvic tissue other than uterus and tube”, and “tumor involvement of the small bowel and mesentery” are directly associated with FIGO stage. Especially, the latter is also associated with residual tumor size as multiple tumors in the small bowel and mesentery are difficult to remove completely. Putting together, the combination of the selected variables seems to be more suitable for reflecting disease status and consequently shows better predictive performance than the FIGO stage and residual tumor size”.

Comment 7: Finally, did the authors not consider including any molecular features for prediction. Given that not all patients are tested for mutations etc., for the patients that do have the available information, would that affect the outcome for patients? This in fact could guide the inclusion of looking at molecular markers in deciding treatment options.
Answer 7: Thank you for your valuable comments. We totally agree with your opinion that molecular features, such as genetic test results, may provide additional information on prognosis and help physicians and patients in deciding treatment options. Recently, investigation of HGSOC patients’ germline or somatic BRCA1/2 mutation status emerged owing to the advent of several PARP inhibitors in management of ovarian cancer. However, in this study, information on each individual’s BRCA1/2 mutational status were limited: more than a half of patients did not undergo any type of gene tests. Such low proportion of HGSOC patients who underwent BRCA1/2 gene testing might originate from the fact that it was not until October 2019 that the Korean FDA permitted olaparib maintenance therapy in patients with BRCAmut, primary high-grade ovarian cancer. We believe that incorporation of specific gene test results or molecular markers, which you suggested, will improve predictive performance of models, considerably. Indeed, such studies are definitely warranted in near future. But at this point, we are very sorry that we have no choice but to add these points to the limitations of the current study. Please understand our efforts generously.
(Page 12, Lines 396-401) “Third, molecular features, such as specific gene mutations, were not considered in this study. Regarding BRCA1/2 mutation status, only a small portion of the patients underwent germline or somatic BRCA1/2 gene testing in our institutions. By incorporating BRRCA1/2 gene test results and other genetic alterations, we believe that it would be possible to develop predictive models with higher accuracy”.

Thank you for your consideration. We look forward to hearing from you.
(Ref.)
[1] Bowtell
DD, Böhm S, Ahmed AA, Aspuria PJ, Bast RC, Jr., Beral V, et al. Rethinking ovarian cancer II: reducing mortality from
high grade serous ovarian cancer. Nature reviews Cancer. 2015;15:668 79.
[2] Integrated genomic analyses of ovarian carcinoma. Nature. 201
1;474:609 15.
[3] Zhang H, Liu T, Zhang Z, Payne SH, Zhang B, McDermott JE, et al. Integrated Proteogenomic Characterization of Human High
Grade Serous Ovarian Cancer. Cell. 2016;166:755 65.
[4] Kim SI, Song M, Hwangbo S, Lee S, Cho U, Kim JH, et al.
Development of Web Based Nomograms to Predict Treatment Response
and Prognosis of Epithelial Ovarian Cancer. Cancer research and treatment. 2019;51:1144 55.
[5] Yongkang Kim M
SK, Yonghwan Choi, Sung Gon Yi, Junghyun Namkung, Sangjo Han, Wooil Kwon, Sun Wh e Kim Jin Young
Jang, Hyunsoo Kim, Youngsoo Kim, Seungyeoun Lee, Taesung Park. Comparative studies for developing protein based cancer
prediction model to maximise the ROC AUC with various variable selection methods. International Journal of Data Mining an d
Bioinformatics.16:64 76.

Round 2

Reviewer 1 Report

The authors responded well to the comments and improved their manuscript. Except some minor remarks the manuscript is acceptable for publication in Cancers.

Minor remarks:

  1. In Table S4 the authors state the fitted results of the logistic regression model used for nomogram development. However, they included FIGO stage and residual tumor in Table S4- two parameters which are not included in the final LR model. Please clarify.
  2. In the first review the inclusion of the used cut-off for sensitivity/specificity calculations was requested (comment 4). The authors included the cut-off for the LR model. However, the authors should translate this cut-off into the nomogram score (total points) and state this cut-off.

Author Response

Please see the attachment for details, thanks.

Reviewer 1
The authors responded well to the comments and improved their manuscript. Except some minor remarks the manuscript is acceptable for publication in Cancers.

Comment 1:
In Table S4 the authors state the fitted results of the logistic regression model used for nomogram development. However, they included FIGO stage and residual tumor in Table S4 two parameters, which are not included in the final LR model. Please clarify.
Answer 1:
Thank you for your thorough review and words of encouragement. The reviewer’s comment is correct: we are very sorry for our mistake. The original Table S4 was for the 8-variable LR model. Preparing this revision, we changed Table S4 to those for the final 6-variable LR model as follows:

Table S4 . Fitted results of the logistic regression model used for nomogram development.

Comment 2:
In the first review the inclusion of the used cut off for sensitivity/specificity calculations was requested (comment 4). The authors included the cut off for the LR model. However, the authors should translate this cut off into the nomogram score (total points) and state this cut off.

Answer 2:
We appreciate your valuable comments.
In response to the reviewer’s comment, we translated the cut
off into the nomogram score. The cut-off value of 0.175 for the six variable LR based model corresponded to the 41 point s for the nomogram. Our n omogram shows total points as well as the probability of being platinum resistant cases. We regarded the case with total points >=41 as a high risk group.

As per the reviewer’s comments, we rs comments, we revised Figure 4, which now presents information on the cut information on the cut--off point off point and which group (high vs. low) an individual belongs to.

We also the following sentence in Results section:

(Page 8) The developed nomogram presents total points as well as the probability of being platinum resistant cases. Based on the cut
off value of 41 points, we regarded the case with total point ≥41 as a high risk group

Thank you for your consideration. We look forward to hearing from you.
